cellular biology/ecology/physiology

telomeres, blood cell type, nucleated versus non-nucleated, lizard, *Ctenophorus pictus*

**Author for correspondence:**
Mats Olsson
e-mail: mats.olsson@bioenv.gu.se

# Telomere length varies substantially between blood cell types in a reptile

Mats Olsson[1,2], Nicholas J. Geraghty[2], Erik Wapstra[4] and Mark Wilson[3]

[1]Department of Biological and Environmental Sciences, University of Gothenburg, Gothenburg, Sweden
[2]The School of Earth, Atmospheric and Life Sciences, and [3]School of Chemistry and Molecular Bioscience, Illawarra Health and Medical Research Institute, University of Wollongong, Australia
[4]School of Natural Sciences, University of Tasmania, Hobart, Australia

MO, 0000-0002-4130-1323

Telomeres are repeat sequences of non-coding DNA-protein molecules that cap or intersperse metazoan chromosomes. Interest in telomeres has increased exponentially in recent years, to now include their ongoing dynamics and evolution within natural populations where individuals vary in telomere attributes. Phylogenetic analyses show profound differences in telomere length across non-model taxa. However, telomeres may also differ in length within individuals and between tissues. The latter becomes a potential source of error when researchers use different tissues for extracting DNA for telomere analysis and scientific inference. A commonly used tissue type for assessing telomere length is blood, a tissue that itself varies in terms of nuclear content among taxa, in particular to what degree their thrombocytes and red blood cells (RBCs) contain nuclei or not. Specifically, when RBCs lack nuclei, leucocytes become the main source of telomeric DNA. RBCs and leucocytes differ in lifespan and how long they have been exposed to 'senescence' and erosion effects. We report on a study in which cells in whole blood from individual Australian painted dragon lizards (*Ctenophorus pictus*) were identified using flow cytometry and their telomere length simultaneously measured. Lymphocyte telomeres were on average 270% longer than RBC telomeres, and in azurophils (a reptilian monocyte), telomeres were more than 388% longer than those in RBCs. If this variation in telomere length among different blood cell types is a widespread phenomenon, and DNA for comparative telomere analyses are sourced from whole blood, evolutionary inference of telomere traits among taxa may be seriously complicated by the blood cell type comprising the main source of DNA.

# 1. Introduction

The number of studies of telomeres, the non-coding protein-nucleotide 'caps' primarily at the ends of chromosomes, has rapidly increased recently across a wide range of disciplines, from functional mechanisms and biochemistry to their ongoing evolution in the wild [1–5]. The conserved, non-coding protein-(TTAGGG)$_n$ sequences in most metazoans [6] protect the chromosome ends from erosion and from the cell's own DNA repair system but suffer from attrition during cell replication and onslaught by reactive molecules [1–3]. Telomere sequences interspersed within the chromosomes themselves (interstitial telomeres) are likely to be more stable over time but may have negative effects on chromosomal stability and risk of genetic disease, such as cancer [1–5]. The most important regulator of telomere length and net attrition rate is the telomerase enzyme, but other repair systems also play major roles in telomere dynamics [7,8]. Importantly, key telomere traits, such as length and rate of attrition, are linked to a range of phenotypic traits including risk of cancer [9] and other forms of genetic disease [10–12], diabetes [12], infections [10,12], metabolic rate [1,2,4,5] and lifespan [1,2], and may differ between gender [13,14], ethnic groups [12], morphs within the same species [15] and phylogenetically different taxa [6]. In addition, telomere heritability has been estimated as having widely different values, from zero to more than one (reviewed in [4]), which makes inferences of their ongoing evolution hard to depict.

Given telomere links and correlations to profoundly important biomedical traits, understanding telomeres' fundamental role(s) in biology critically depends on uniformity in procedures across compared groups, whether telomeres are just 'score sheets' of other processes dictating probability of viability and fitness, or are themselves causally predicting fitness. Researchers have therefore assessed congruence in telomere measurements across different tissue types, finding that sometimes telomeres measured in blood cells are longer than those measured in spleen, sperm, liver, brain, heart and muscle (e.g. [16,17]); however, in other cases, all tissue types appear to have corresponding telomere lengths (e.g. [18–20]). While acknowledging the importance of understanding differences in telomere biology between tissue types that require destructive sampling of the organism, here we specifically address issues with differences in a tissue type that allows repeat sampling of individual organisms through life. Most commonly, blood is used for such purposes, since this involves less invasive sampling. Importantly, however, some blood cells only occur in some taxa. For example, only squamate reptiles have azurophils (a monocyte involved in phagocytosis; [21]), and mammals lack nucleated erythrocytes. Furthermore, blood cell types typically develop along two different pathways with significant implications for telomere attrition: (i) myeloid type cells develop into erythrocytes and thrombocytes, in addition to short-lived innate leucocytes (e.g. granulocytes and monocytes). These cells do not divide once split off from stem cells. (ii) Lymphoid cell types develop into longer lived T and B cells and these, however, have the capacity to proliferate with potential effects for telomere base pair loss (end replication problem; [22]). In addition, the lifespan of different blood cell types can sometimes vary by orders of magnitudes across taxa [23]. In mammals, red blood cells can live for approximately 120 days and leucocytes on average live for approximately 12–13 days [24], but in reptiles, red blood cells can live for 600–800 days [21], while the lifespan of white cell types are to the best of our knowledge unknown (no hits in wide literature searches). Even if (most) of these cell types do not go through mitosis once generated from haematopoietic stem cells, telomere attrition rates could vary profoundly, depending on exposure time to reactive oxygen species (ROS) and other reactive molecules that affect attrition [5,25]. Therefore, to assess to what extent different cell types in the same individual and blood sample have different telomere length, we set out to identify blood cells of different types using flow cytometry and simultaneously measure the lengths of their telomeres using the same technique [26–28].

# 2. Material and methods

## 2.1. Field study and blood sampling

All field protocols have been published elsewhere (see [29]), and we therefore only give a brief summary here. Nineteen adult male and 18 female painted dragon lizards (*Ctenophorus pictus*) less than one-year old were caught at Yathong Nature Reserve, NSW, Australia (145°35′ E, 32°35′ S) during Spring 2018. Approximately 50 µl whole blood was taken in the corner of the mouth by rupturing *vena angularis* with a needle after which the blood was collected in a capillary tube. Aliquots (10 µl) of each sample

were immediately diluted into 1.5 ml of cryopreservation buffer (30% v/v fetal bovine serum (Sigma 12003C), 60% v/v RPMI-1640 medium (Sigma R8758) and 10% v/v DMSO (Sigma D4540)) for flow cytometric analysis.

In painted dragons, males are polymorphic with respect to head and bib colour which phenotypically reveals differences in life-history strategies, including differences in somatic self-maintenance, reproductive expenditure and telomere attrition [30]. Males have red, orange, yellow or 'blue' heads (no yellow/red pigmentation) and most research has focused on the red and yellow morphs as they have been more common than orange and blue morphs in recent history [31]. Yellow males have larger testes and have nearly four times as high reproductive success in sperm competition trials than red males, despite shorter copulations [32]. Thus, these morphs show alternative reproductive tactics reflecting differences in trade-offs of resources between testes, testosterone-driven aggression and longevity. Additionally, males with a gular bib are more reproductively successful, producing more single paternity clutches, while suffering greater loss in body condition due to mate defence than males without bibs [32]. The variation in reproduction and self-maintenance trade-offs among morphs probably stems from differences at a cellular level [15,33].

## 2.2. Microscopy and cell identification

Fixed blood samples were kept at 4°C until use. Blood was stained with 4′,6-diamidino-2-phenylindole (DAPI) for 15 min in the dark. Blood was then aliquoted onto pathology slides (Livingstone, Mascot, Australia) and mounted by addition of Citifluor AF100 + BVCVOF as per manufacturer's instructions (Electron Microscopy Sciences Group, Hatfield, PA, USA). Images were captured and analysed using a Leica SP8 confocal microscope (Wetzler, Germany) and Leica Application Suite X software (version 3.4.2). Cell types were identified based on size, cellular morphology and comparison to published Giemsa stained cells [27,28,34–36]. Following this literature, we conservatively identified three cell types, red blood cells (RBCs), lymphocytes (T or B cells, not differentiated) and the reptile-specific azurophils (a reptilian monocyte; [21]).

## 2.3. Flow FISH analysis—quantifying telomere length

We followed manufacturer guidelines and published species-specific methods (see [29]). In brief, standard cryopreservation methods for blood cells were used [37]. Haemocytometer counts determined that *C. pictus* whole blood contains approximately $1 \times 10^9$ RBCs ml$^{-1}$; therefore, 10 µl of whole blood from *C. pictus* contains $6 \times 10^6$ cells, which is three times the number of cells required for the FISH (fluorescence *in situ* hybridization) analysis. Whole blood (10 µl) was added to 1.5 ml cryopreservation buffer kept on ice to achieve the recommended cell mass of $2–4 \times 10^6$ per ml of cryo-buffer. Samples were placed in a foam box in a −80°C freezer to slowly cool the cells to avoid rupture, where they were kept until analysis.

We used the Telomere PNA Kit/FITC for flow cytometry (Dako), recommended by the manufacturer for use with nucleated cells from all vertebrates (http://www.dako.com/au/ar42/p107840/prod_products. htm). The kit is based upon the hybridization of a synthetic DNA/RNA analogue, conjugated with fluorescein isothiocyanate (FITC), capable of binding to telomeres in a sequence-specific manner obeying the Watson–Crick base pairing rules. This kit does not suffer from the interaction of sub-telomeric sequences, unlike methods such as telomere restriction fragment analysis using southern blotting (TRF), and the probe hybridizes with telomere repeat sequences (TTAGGG) typical of vertebrates including lizards. The resulting fluorescence intensity of the cells is directly correlated with the length of the telomeres [26–28]. This method therefore provides a relative indication of telomere length between different cells. Tests of reproducibility of the Telomere PNA kit/FITC were performed at Dako's laboratories using human blood [29]. Relative telomere length-values showed a coefficient of variation of 8–13% of single determinations and 6–9% for duplicate determinations (see [29])

Cryopreserved blood was brought to room temperature and washed in PBS (135 mM NaCl, 2.7 mM KCl, 1.75 mM KH$_2$PO$_4$, 10 mM Na$_2$HPO$_4$, pH 7.4) by centrifugation (as described above), before subsequent processing within 4 h of sampling. Relative telomere length was compared between blood cells of individual lizards using the Dako kit, following the manufacturer's instructions. Cells were counter-stained with propidium iodide (PI) to assess total nucleic acid content. Flow cytometry was performed using a Becton Dickinson Fortessa X-20, with excitation at 488 nm and emitted fluorescence collected using band pass filters of 515 ± 10 nm (fluorescein) and 695 ± 10 nm (PI). More than 90% of the cells had a similar level of PI staining and this major population was electronically gated to select them for analysis of the level of telomere probe hybridization. Between 5 and 9% of blood cells had

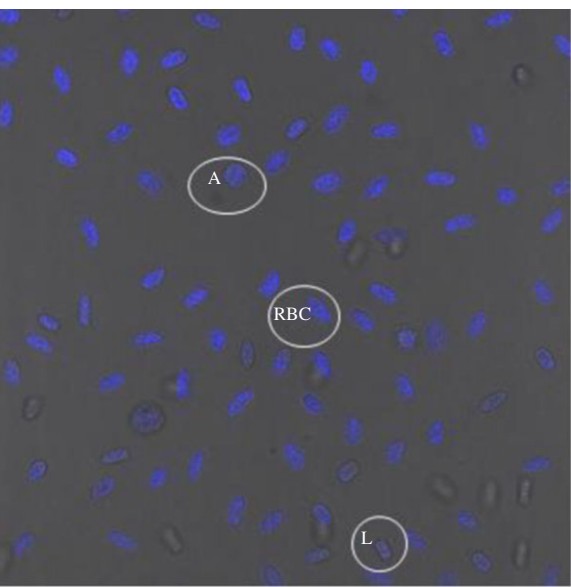

**Figure 1.** A representative confocal microscopy image of DAPI stained blood cells of *Ctenophorus pictus*, with azurophil (A), lymphocyte (L), and red blood cell (RBC) highlighted.

higher levels of PI staining and these were electronically excluded from the analyses (the excluded lizard blood cells may be undergoing cell division and therefore contain inherently higher levels of nucleic acids—see also [29]). Cultured CCRF-CEM cells (human acute lymphoblastic leukaemic cell line; Sigma) were used as control cells and were mixed with each blood sample before processing and analysis. For each individual sample, the telomere-associated FITC fluorescence of lizard blood cells was reported relative to that of an internal standard (co-processed and co-analysed control CCRF-CEM cells). Cell types were identified based on size using forward scatter area (FSC-A) and cellular morphology (specifically granularity; figure 1) using side scatter area (SSC-A) with excitation at 488 nm and comparison to previously published identifications ([27,35]; figure 2).

## 2.4. Statistical methods

The telomere distributions of all three cell types examined (azurophils, RBCs and lymphocytes) were slightly left-skewed and only the distribution of azurophil telomeres could be normalized by log transformation. We therefore used non-parametric statistics in all comparisons between groups (Kruskal–Wallis tests). We used *t*-tests supported by Kruskal–Wallis tests to look for differences in telomere length between RBCs and lymphocytes because this comparison is more easily generalized to other taxa, that is, we excluded azurophils as these only occur in reptiles (but see [21]). For multiple regression analyses with telomere lengths of different cell types as response variable, we used generalized linear models (PROC GLM in SAS 9.4), including male morphs and body size, and separately analysing sex effects (since females are largely uniformly brownish-grey and camouflaged).

## 3. Results

We first examined the overall differences in telomere lengths across the three cell types and all measurements (ignoring sex, size and male morphs) showed highly significant differences in telomere length among cell types. Azurophils had the longest telomeres, RBCs the shortest and lymphocyte telomeres were intermediate in length (figure 3; $X^2 = 76.9$, $p < 0.0001$, $N = 37$ for each cell sample). With azurophils excluded, a *t*-test revealed that mean telomere length in lymphocytes was nearly three times the length of telomeres in RBCs (270%; mean telomere length ± s.d.: lymphocytes $0.26 \pm 0.124$, RBCs $0.09 \pm 0.045$; $t = -7.50$, $DF_{\text{Satterthwaite}} = 45.3$, $p < 0.0001$). A Kruskal–Wallis test confirmed the *t*-test ($X^2 = 48.7$, $p < 0.0001$). The corresponding comparison between azurophil and RBC telomeres showed an even greater difference (388%; mean azurophil telomere length $= 0.37 \pm 0.131$, $t = 11.97$, $p < 0.0001$, confirmed by the Kruskal–Wallis test $X^2 = 52.5$, $p < 0.0001$). The difference in telomere length between lymphocytes and azurophils was also statistically significant (143%; $t = 3.73$, $p = 0.0004$;

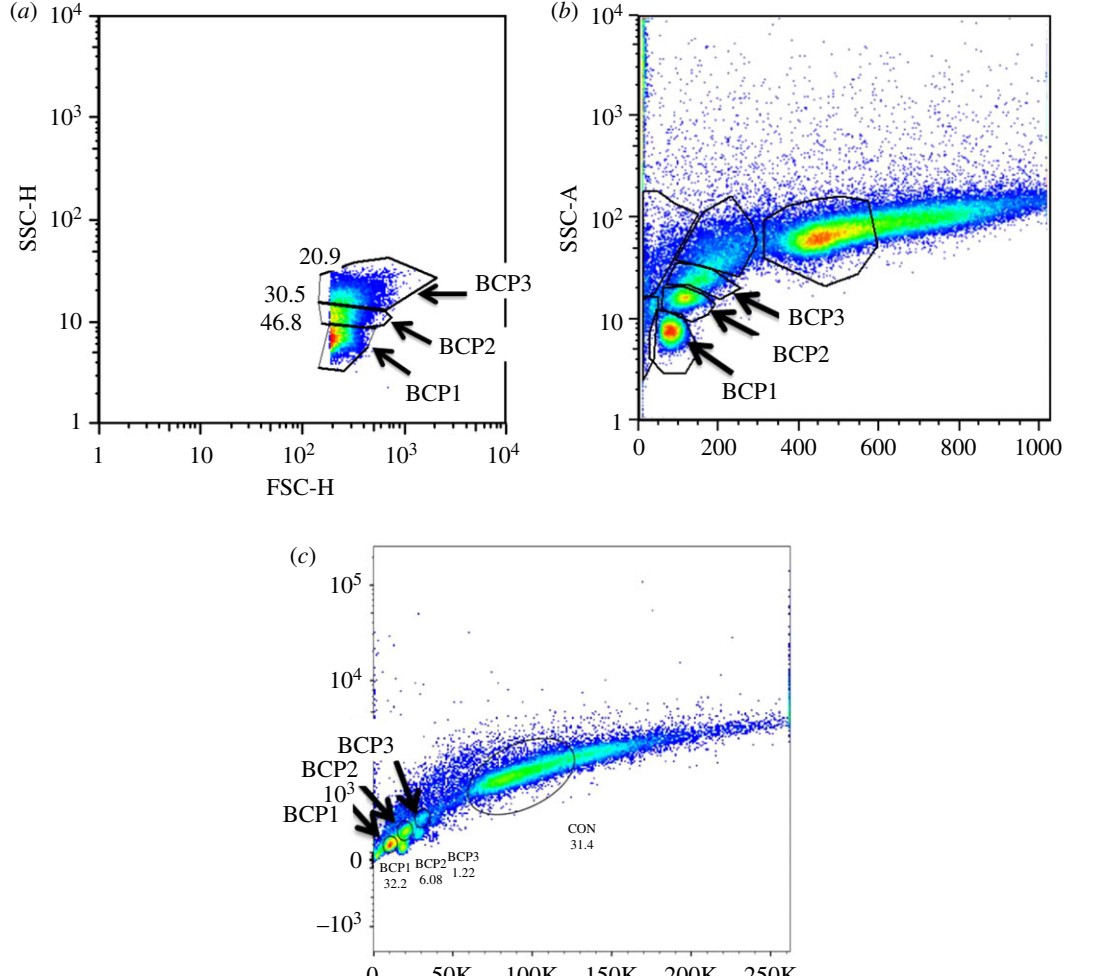

**Figure 2.** Cell types of *Ctenophorus pictus*, designated as blood cell population (BCP) 1, 2 and 3, were identified (*a*) based on size (FSC-H) and cellular morphology (SSC-H) and (*b*) SSC-A and (*c*) nucleic acid content (using PI; [488] 695/40-A).

Kruskal–Wallis test, $X^2 = 19.3$, $p < 0.0001$). Finally, we looked for correlations between the telomere lengths of the three cell types, which showed that telomeres in RBCs and lymphocytes were correlated in length with each other ($r_s = 0.59$, $p = 0.0001$), whereas neither RBC nor lymphocyte length was correlated with the telomere length of azurophils (with RBS telomeres $r_s = -0.18$, $p = 0.28$, with lymphocytes $r_s = 0.20$, $p = 0.23$).

We then analysed telomere length for each cell type with sex and body size (mass) as predictors in generalized linear models. There were no effects of sex, mass or their interactions on RBC telomere length (model $F_{3,32} = 1.06$, $p = 0.38$, $R^2 = 0.09$). For lymphocytes, however, with the non-significant interaction backwards-eliminated ($p = 0.46$), female telomere length was longer than for males (parameter estimate = 0.13, $t = 3.15$, $p = 0.0034$; model $F_{2,33} = 4.97$, $p = 0.013$, $R^2 = 0.23$). For azurophils, with the non-significant interaction removed ($p = 0.81$), there was a borderline negative effect of body mass (parameter estimate = $-0.029 \pm 0.014$, $p = 0.056$; model $F_{2,33} = 5.25$, $p = 0.010$), but with no significant sex effect ($p = 0.13$).

We then analysed the effects of male head colour morph and the presence and absence of bibs on the telomere length of each blood cell type. None of these effects was significant and will not be further reported on ($p > 0.21$ in all analyses).

## 4. Discussion

Our results show clear effects of blood cell type on the length of telomeres in simultaneously sampled individuals using identical protocols. The magnitudes of differences in average cell type-specific

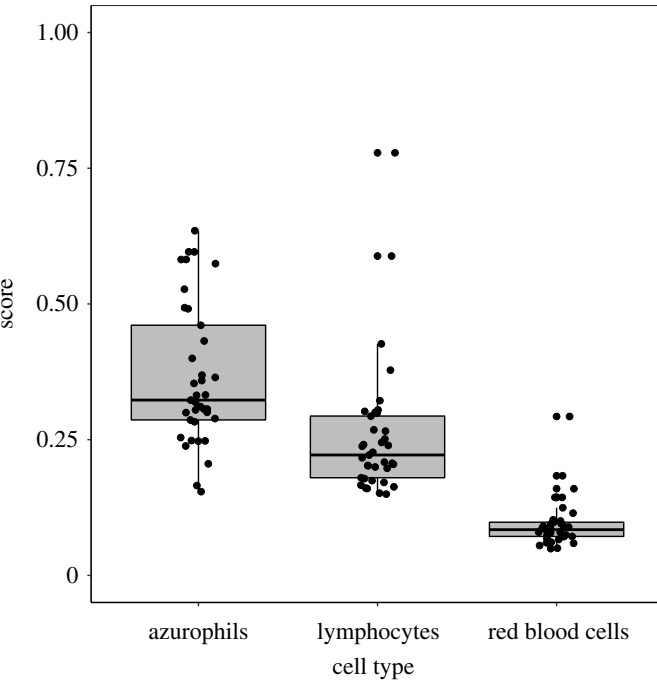

**Figure 3.** Box plot of rank sums of telomere lengths for azurophils, lymphocytes, and red blood cells from *Ctenophorus pictus*. The Wilcoxon scores are significantly different among groups ($X^2 = 76.9$, $p < 0.0001$, $n = 37$ for each cell sample).

telomere length varied among cell types, with lymphocytes and azurophils having approximately 2–3 times longer telomeres than those of red blood cells. This could potentially strongly influence a number of comparative analyses that use, or are constrained to use, different cell types to generate DNA and telomeric data. Across metazoans, telomere length differs by many orders of magnitude, from less than 1 kb in some Ctenophoras (*Pleurobrachia pileus*) to some 3 Mb in Class III telomeres in chickens, thus being some 3000 times longer (*Gallus gallus*) [6]. Thus, in some cases, the relative differences among taxa may be large enough not to be masked by differences in blood cell sources of telomere DNA. In other cases, however, telomere length differences of a factor up to four times may create spurious relationships (Type I error, false positives), and other relationships could be missed in data noise and thus be overlooked (Type II error, false negatives). One such group of comparisons would be particularly likely to be prone to error—that between mammals (with leucocytes being the only source of telomeric DNA) and birds and ectotherms (in which erythrocytes constitute more than 95% of nucleated blood cells and presumably a corresponding proportion of telomeric DNA; [24]).

In our previous study of painted dragons [17], we found that telomeres were longer in whole blood cells than in the brain, heart and spleen. Little is known about the blood telomeres of reptiles generally, which may simply be longer than telomeres of other tissue types. An alternative scenario is that these comparisons may be influenced by methodological differences in DNA extraction (or similar); to fully lyse cells in organ samples takes substantially longer time and may cause more damage to the telomeres than to the qPCR reference gene due to their different locations on the chromosomes. This could potentially result in shorter telomere measurements. Importantly, in our current study of blood cell type differences in telomere length, there are no differences in methodology and treatment of the compared cell type groups—they are all treated with the same sampling methods and flow cytometry protocol.

Given the differences in developmental pathways and subsequent proliferation rate between myeloid and lymphoid cells, there are differences in the number of mitotic steps that could cause differences in telomere attrition simply from nucleotide loss at cell fission [1]. Several other processes may also contribute to blood cell-specific telomere length, their attrition and the mean telomere length resulting from averaging over large populations of cells—different in types or not. Here, we discuss potential such modifiers of blood cell-specific telomere length and their average measure across cells:

(i) Telomere length 'from start'. Differences in telomere length are set at the earliest stages of cell life. Only 25% of cells in active marrow belong to the erythrocyte-producing series compared to 75%

that mature into leucocytes (even though there are 500 times more circulating RBCs than leucocytes in humans; [24]). Depending on telomere length in different pluripotent, uncommitted haematopoietic stem cells, telomere length varies among chromosomes in blood cells [10] and are for a number of reasons outside the scope of this study (e.g. [1,4,5,38]).

(ii) Blood cell lifespan. In mammals, an erythrocyte's lifespan is approximately 120 days and takes 5–7 days to produce in the bone marrow [24]. Leucocyte lifespan, however, varies considerably from minutes (at acute infection) to approximately 12 days [24]. We have found no published literature on the lifespan of leucocytes in squamate (or other) reptiles but in another ectotherm—the ginbuna crucian carp (*Carassius auratus langsdorfii*), RBCs have a half-life of approximately 51 days on average (with a small fraction living for up to 270 days; [39]). The half-life of circulating lymphocytes ranged from 10 to 23 days, while a fraction of lymphocytes (probably memory cells) remained in circulation for up to 145 days. The corresponding half-life of monocytes was 2–3 days and 2–27 days for granulocytes [39]. Assuming the same approximate relations apply to painted dragons, and the fact that reptilian erythrocytes have a lifespan of 600–800 days [39], the majority of erythrocytes should thus be exposed to onslaught by reactive molecules for much longer, perhaps several hundred per cent longer, than lymphocytes and other leucocytes.

(iii) Proportions of cells in a 'mixed cell population average'. Given the above scenario, it should be obvious that averaging over large populations of cells consisting of a mix of blood cell types will generate different mean telomere lengths. So what are representative cell type proportions and how do they vary? The best data on cell compositions come from humans: neutrophils, 50–70% of leucocytes, lifespan around 10 h; eosinophils, 2–4% of leucocytes, a half-life of 18 h and a lifespan of about a week; basophils, 1% of leucocytes, lifespan up to 24 h; monocytes, 2–8% of leucocytes, lifespan 24 h; lymphocytes, 20–30% of leucocytes, lifespan a week to a few months; erythrocytes, 500 times more common than all leucocytes put together, and a lifespan of approximately 120 days [24]. The relative proportions of these cell types are strongly susceptible to infections [40], other disease [4,10] and a plethora of other factors. In reptiles, lymphocyte counts vary between species and can constitute as much as 80% of all leucocytes in some species [21], and vary in proportion over seasons, being lower in winter and higher in summer in many ectotherms [21]. Lymphopenia (low level of lymphocytes) covaries negatively with malnutrition and stress [21]. By contrast, lymphocytosis (of course) covaries with wound healing, inflammatory disease, parasitic infections and viral disease [21].

(iv) ROS regulation and DNA repair of telomeres. Telomeres are prone to damage from metabolic processes, which results in complex interactions between reactive molecules, oxidative stress and telomere shortening [3,29,41]. A contributing factor to such telomere attrition under oxidative stress is a decrease in telomerase activity [42]. Thus, at a cellular level, a number of dynamic regulatory processes affect telomere attrition and subsequent repair [25,43]. We have previously shown significant (slight) effects of DNA repair on telomere elongation, which supports the importance of DNA repair systems, in addition to telomerase, for restoring telomere length [29]. Once the telomeric sequence is repaired, you would expect regained ability to hybridize, and, hence, more sequence repeats identified by Flow FISH (i.e. longer telomeres).

From this perspective, telomere shortening comes about through inadequately repaired double- or single-strand breaks [25]. There is, however, an additional route through which ROS could affect telomere length—by direct intervention with haematopoiesis [44]. Recent work shows that ROS could be crucial in the regulation of the balance between self-renewal and differentiation of haematopoietic stem cells (HSC; [44]). The mechanistic details in these processes are outside the scope of this report (see details in [44]). It suffices to say that given ROS' ability to dictate HSC lifespan and regeneration, their differentiation (i.e. relative allocation into different cell types of pluripotent stem cells and progenitor cells) and shorter or longer term blood cell survival that averaging over these cell types will give different telomere length means.

In summary, our study reveals that our model species, Australian painted dragon lizards (*Ctenophorus pictus*), have considerably longer telomeres in their lymphocytes and azurophils than in their red blood cells. If this is a general phenomenon, it complicates comparative analyses across taxa with and without nucleated red blood cells. Averaging telomere length over these cell types will thus generate differences in mean telomere length and proportional shifts in cell types may explain differences in telomere length between taxa with and without nucleated RBCs, differences in health status, and perhaps explain drastic shifts in telomere lengths that have previously been hard to explain. This encourages careful scrutiny of

haematological factors underpinning telomere lengths, in particular in comparative studies across taxa with species with different blood cell type compositions.

Ethics. This work was performed under the Animal Ethics permit AE17/27 at the University of Wollongong and permits from NSW Parks and Wildlife Service (SL 100352).

Data accessibility. Data are available in the electronic supplementary material.

Authors' contributions. All authors made substantial contributions to this study including in its design, data collection and interpretation. Specifically, M.O. and M.W. conceived the ideas, and the fieldwork was carried out by M.O. and laboratory analyses by N.J.G. and M.W. M.O. led the writing of the manuscript with contributions by all authors. All authors contributed to the drafting and revising of the article and approved the final version.

Competing interests. The authors declare they have no competing interests.

Funding. This project was (partially) funded by the Australian Research Grant DP140104454.

Acknowledgements. The authors were funded by the Australian Research Council (DP140104454). We thank three anonymous reviewers and the editors for helpful suggestions on the manuscript.

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
