## [Reviewer comments · Royal Society Open Science]

Review History

RSOS-192136.R0 (Original submission)

Review form: Reviewer 1 (Mar Comas)

Is the manuscript scientifically sound in its present form?

Yes

Are the interpretations and conclusions justified by the results?

Yes

Is the language acceptable?

Yes

Do you have any ethical concerns with this paper?

No

Have you any concerns about statistical analyses in this paper?

No

Recommendation?

Accept with minor revision (please list in comments)

Comments to the Author(s)

Dear Dr. Olsson and coauthors,

I have read and reviewed the manuscript: "Telomere length varies substantially between blood cell types in a reptile" (RSOS-192136).

I think your manuscript is interesting, well written and worth to publish it. This manuscript apports new and valuable information that should be taken into account in future studies done with telomeres, especially those done from blood samples.

My only general concern is about methods which, from my point of view, are briefly explained and more details may be required. I want to point that there are references cited, but some more information may be useful for a general reader, not interested in review all information referent to telomeres already published. Moreover, a more detailed methodology about flow cytometry may be useful.

I made a few specific comments below.

Line 117: please, after blue heads close the "

Line 158: please, provide a reference or data supporting the sentence: "The resulting fluorescence intensity of the cells is directly correlated with the length of the telomeres"

Line 159: You provide information on the reproducibility of the techniques, but it may be interesting also provide a measurement of repeatability of your own data.

Line 176: please, delete one r of forr.

Mar Comas

Review form: Reviewer 2

Is the manuscript scientifically sound in its present form?

Yes

Are the interpretations and conclusions justified by the results?

Yes

Is the language acceptable?

Yes

Do you have any ethical concerns with this paper?

No

Have you any concerns about statistical analyses in this paper?

No

Recommendation?

Accept with minor revision (please list in comments)

Comments to the Author(s)

Dear Authors,

I have enjoyed reading your manuscript. It discuss a very common issue in measuring telomere length from blood. At any given time point blood could vary substantially in promotion of different blood cells and how will this affect the average telomere length of blood need to be carefully investigated in different species. Manuscript is written well and I have a few comments that will help you to revise the manuscript.

1. why authors did not include the age in the model, which is the strongest predictor of telomere length in blood.
2. How telomere length of these three cell types correlates to blood telomere length. Why authors did not compare it.
3. Discussion need shorten and more focused. There is data available in mammals investigating the telomere length in different cell lines as well as it correlation.
4. Line 179, correct the Typo, Forr
5. Line 290-291, does author means Asghar et al. 2018?
6. line 296 - 297, Please add reference.
7. Line 327-330. In recent study, it has been shown that although cell promotion change during infection but does not correlate with whole blood telomere length (Asghar et al. 2018).

Review form: Reviewer 3

Is the manuscript scientifically sound in its present form?

Yes

Are the interpretations and conclusions justified by the results?

Yes

Is the language acceptable?

Yes

Do you have any ethical concerns with this paper?

No

Have you any concerns about statistical analyses in this paper?

Yes

Recommendation?

Accept with minor revision (please list in comments)

Comments to the Author(s)

This is a really interesting study on average telomere length differences in three blood cells types in reptiles: red blood cells, lymphocytes and azurophils. The authors use a study system, the painted dragon, in which they have plenty of experience and information, in addition to having developed a robust methodology for their aim.

I generally agree on their interpretation of the results, though I feel the authors should take more careful consideration when stressing the implications of their findings in the evolutionary context.

I agree on the importance of the extant differences in telomere length between cell types. However, I feel the discussion of the authors on this respect could be more focus as I find it too loose for the brief of the results, i.e. differences on average telomere length among three blood cell types.

Furthermore, as stated by the authors (L282-297), the proportion of each cell type should be critical to over/ underestimate blood cell average telomere length. Indeed, in the study model, the longer average telomere length in the azurophils might not be relevant for the final telomere length of the individual. I wonder if the authors have information on the final telomere length estimate in the whole blood sample and in that case how the estimated within-cell telomere length influence this? I think adding this kind of information will provide a direct example for the point discussed in L282-297.

I understand that cell counts are not always easily normalised, but have the authors considered to analyse the telomere distribution of these three types in a single model instead (L184-187)? E.g. using the current DataBase in long-format with telomere length as a dependent variable, the cell type as a 3-level factor and the ID of the individual as a random effect to account for the non-independence in the measurements.

Minor

L104-126. I suggest to directly provide in the text the information about the individuals used in the study. It is nice to have information about the species ecology but sex, age or sample size should be easily accessible, so far only available in the DataBase.

L190-193. Could the authors be slightly more specific about the model selection process used? Also, the authors state that they used mixed models, in that case which is the random structure? Please state it.

Figure 3. Please explain in the figure legend the x-axis or reword it. I find the current nomenclature confusing as it might be confused by the proportion of total cells. Also there is a potential typo, the two versions of the image have different axis title and scale.

Decision letter (RSOS-192136.R0)

18-Mar-2020

Dear Professor Olsson

On behalf of the Editors, I am pleased to inform you that your Manuscript RSOS-192136 entitled "Telomere length varies substantially between blood cell types in a reptile" has been accepted for publication in Royal Society Open Science subject to minor revision in accordance with the referee suggestions. Please find the referees' comments at the end of this email.

The reviewers and handling editors have recommended publication, but also suggest some minor revisions to your manuscript. Therefore, I invite you to respond to the comments and revise your manuscript.

- Ethics statement

- Data accessibility

<http://datadryad.org/submit?journalID=RSOS&manu=RSOS-192136>

- Competing interests

- Authors' contributions

- Acknowledgements

- Funding statement

Because the schedule for publication is very tight, it is a condition of publication that you submit

the revised version of your manuscript before 27-Mar-2020. Please note that the revision deadline will expire at 00.00am on this date. If you do not think you will be able to meet this date please let me know immediately.

If your manuscript is newly submitted and subsequently accepted for publication, you will be asked to pay the article processing charge, unless you request a waiver and this is approved by Royal Society Publishing. You can find out more about the charges at

<https://royalsocietypublishing.org/rsos/charges>. Should you have any queries, please contact openscience@royalsociety.org.

Kind regards,

Anita Kristiansen
Editorial Coordinator

on behalf of Kevin Padian (Subject Editor)
openscience@royalsociety.org

Associate Editor Comments to Author:

Comments to the Author:

Thank you for the submission, which has been reviewed by three referees. They all recommend that, after a number of comparatively minor modifications, your paper may be accepted. Please ensure you engage constructively with their comments, and provide a full point-by-point response in addition to a revised manuscript. We'll look forward to receiving the updated paper in due course.

Reviewer comments to Author:

Reviewer: 1

Comments to the Author(s)

Dear Dr. Olsson and coauthors,

I have read and reviewed the manuscript: "Telomere length varies substantially between blood cell types in a reptile" (RSOS-192136).

I think your manuscript is interesting, well written and worth to publish it. This manuscript apports new and valuable information that should be taken into account in future studies done with telomeres, especially those done from blood samples.

My only general concern is about methods which, from my point of view, are briefly explained and more details may be required. I want to point that there are references cited, but some more information may be useful for a general reader, not interested in review all information referent to telomeres already published. Moreover, a more detailed methodology about flow cytometry may be useful.

I made a few specific comments below.

Line 117: please, after blue heads close the "

Line 158: please, provide a reference or data supporting the sentence: "The resulting fluorescence intensity of the cells is directly correlated with the length of the telomeres"

Line 159: You provide information on the reproducibility of the techniques, but it may be interesting also provide a measurement of repeatability of your own data.

Line 176: please, delete one r of forr.

Mar Comas

Reviewer: 2

Comments to the Author(s)

Dear Authors,

I have enjoyed reading your manuscript. It discuss a very common issue in measuring telomere length from blood. At any given time point blood could vary substantially in promotion of different blood cells and how will this affect the average telomere length of blood need to be carefully investigated in different species. Manuscript is written well and I have a few comments that will help you to revise the manuscript.

1. why authors did not include the age in the model, which is the strongest predictor of telomere length in blood.
2. How telomere length of these three cell types correlates to blood telomere length. Why authors did not compare it.
3. Discussion need shorten and more focused. There is data available in mammals investigating the telomere length in different cell lines as well as it correlation.
4. Line 179, correct the Typo, Forr
5. Line 290-291, does author means Asghar et al. 2018?
6. line 296 - 297, Please add reference.
7. Line 327-330. In recent study, it has been shown that although cell promotion change during infection but does not correlate with whole blood telomere length (Asghar et al. 2018).

Reviewer: 3

Comments to the Author(s)

This is a really interesting study on average telomere length differences in three blood cells types in reptiles: red blood cells, lymphocytes and azurophils. The authors use a study system, the painted dragon, in which they have plenty of experience and information, in addition to having developed a robust methodology for their aim.

I generally agree on their interpretation of the results, though I feel the authors should take more careful consideration when stressing the implications of their findings in the evolutionary context.

I agree on the importance of the extant differences in telomere length between cell types. However, I feel the discussion of the authors on this respect could be more focus as I find it too loose for the brief of the results, i.e. differences on average telomere length among three blood cell types.

Furthermore, as stated by the authors (L282-297), the proportion of each cell type should be critical to over/ underestimate blood cell average telomere length. Indeed, in the study model, the longer average telomere length in the azurophils might not be relevant for the final telomere length of the individual. I wonder if the authors have information on the final telomere length

estimate in the whole blood sample and in that case how the estimated within-cell telomere length influence this? I think adding this kind of information will provide a direct example for the point discussed in L282-297.

I understand that cell counts are not always easily normalised, but have the authors considered to analyse the telomere distribution of these three types in a single model instead (L184-187)? E.g. using the current DataBase in long-format with telomere length as a dependent variable, the cell type as a 3-level factor and the ID of the individual as a random effect to account for the non-independence in the measurements.

Minor

L104-126. I suggest to directly provide in the text the information about the individuals used in the study. It is nice to have information about the species ecology but sex, age or sample size should be easily accessible, so far only available in the DataBase.

L190-193. Could the authors be slightly more specific about the model selection process used? Also, the authors state that they used mixed models, in that case which is the random structure? Please state it.

Figure 3. Please explain in the figure legend the x-axis or reword it. I find the current nomenclature confusing as it might be confused by the proportion of total cells. Also there is a potential typo, the two versions of the image have different axis title and scale.

Author's Response to Decision Letter for (RSOS-192136.R0)

See Appendix A.

Decision letter (RSOS-192136.R1)

Dear Professor Olsson,

It is a pleasure to accept your manuscript entitled "Telomere length varies substantially between blood cell types in a reptile" in its current form for publication in Royal Society Open Science.

The comments of the reviewer(s) who reviewed your manuscript are included at the foot of this letter.

You can expect to receive a proof of your article in the near future. Please contact the editorial office (openscience_proofs@royalsociety.org) and the production office (openscience@royalsociety.org) to let us know if you are likely to be away from e-mail contact -- if

you are going to be away, please nominate a co-author (if available) to manage the proofing process, and ensure they are copied into your email to the journal.

on behalf of Mr Andrew Dunn (Associate Editor) and Kevin Padian (Subject Editor)
openscience@royalsociety.org

Associate Editor Comments to Author (Mr Andrew Dunn):
Associate Editor
Comments to the Author:
(There are no comments.)

Reviewer comments to Author:

Appendix A

Dear Editors,

Responses to reviewers by MS RSOS-192136 Olsson et al.

Thank you for accepting our MS for publication in RSOS. Please, find below our responses (following **) to the few comments provided by the reviewers:

Reviewer comments to Author:

Reviewer: 1

Comments to the Author(s)

Dear Dr. Olsson and coauthors,

I have read and reviewed the manuscript: "Telomere length varies substantially between blood cell types in a reptile" (RSOS-192136).

I think your manuscript is interesting, well written and worth to publish it. This manuscript apports new and valuable information that should be taken into account in future studies done with telomeres, especially those done from blood samples.

My only general concern is about methods which, from my point of view, are briefly explained and more details may be required. I want to point that there are references cited, but some more information may be useful for a general reader, not interested in review all information referent to telomeres already published. Moreover, a more detailed methodology about flow cytometry may be useful.

** Some of these comments (and the one referring to l. 158 below) seem to suggest that the reviewer is not familiar with Flow Cytometry as a method in general and I don't think this is the right forum for such a review. Cells are incubated with a dye of choice targeting the specific trait to estimate and this results in fluorescence that can be 'counted'. Our methods description on Flow Cytometry (FC) of telomere length, specifically, is already two pages and contains four specific references and the web site from Dako manufacturing the kit. The sentence 'general reader, not interested in review all information referent to telomeres already published' demonstrates further inconsistency between the request for Flow Cytometry methods *per se*, and the huge literature and numerous reviews on telomere methods. We can add additional information if the editor finds this necessary but since this is not a request, but more of a comment, we have left the FC methods as is while adding a reference to line 158 (see below).

I made a few specific comments below.

Line 117: please, after blue heads close the "

** Many thanks, this has been done.

Line 158: please, provide a reference or data supporting the sentence: "The resulting fluorescence intensity of the cells is directly correlated with the length of the telomeres"

** We have cited in (Baerlocher & Lansdorp 2003; Carvalho et al., 2016; 2017) where this is elaborated on - it follows from the hybridization of a synthetic DNA/RNA analogue, conjugated with FITC, binds to telomeres in a sequence-specific manner obeying the Watson-Crick base pairing rules. Each telomere sequence then fluoresces and is counted using standard FC.

Line 159: You provide information on the reproducibility of the techniques, but it may be interesting also provide a measurement of repeatability of your own data.

** The small body size of these lizards (max ca 10g), and the fact that several traits are measured in the same samples, restricts repeatability measures in the same very individuals and samples (see Olsson et al. 2108c more on this). We have therefore in the past assessed repeatability for specific assays separately - this is unfortunately the best we can do and our previous FC analyses show very high correlation coefficients between samples, ranging from 0.80 to 0.97 (Olsson et al. 2008. J Exp Biol 211, 1257-1261.). Dako's own data for the kit used here shows a Relative Telomere Length coefficient of variation of 8-13 % for single determinations and 6-9 % for duplicate determinations.

Line 176: please, delete one r of forr.

** Thank you this has been done.

Mar Comas

Reviewer: 2

Comments to the Author(s)

Dear Authors,

I have enjoyed reading your manuscript. It discuss a very common issue in measuring telomere length from blood. At any given time point blood could vary substantially in promotion of different blood cells and how will this affect the average telomere length of blood need to be carefully investigated in different species. Manuscript is written well and I have a few comments that will help you to revise the manuscript.

1. why authors did not include the age in the model, which is the strongest predictor of telomere length in blood.

** A lot can be said about this but in brief: (i) painted dragons are annuals (ca 10% survive to a second year, but this is probably because they are the ones hatching out the latest the year before). There is very limited possibility to differentiate age between the same cohort of individuals - size could give some indication but there generally is no such effect (see this MS - mass was included in the analyses). (ii) We demonstrate elsewhere (Olsson et al. 2018c) that ROS (DNA damage/repair) is more important determinants of telomere distributions through the year (which we discuss). (iii) Much of the literature of age effects on telomeres are from endotherms, which may have very different somatic telomerase production - so patterns of age-blood telomeres may be uncharacteristic compared to ectothermic taxa.

2. How telomere length of these three cell types correlates to blood telomere length. Why authors did not compare it.

**This is uninterpretable...? The entire MS is about comparing blood cell types...

3. Discussion need shorten and more focused. There is data available in mammals investigating the telomere length in different cell lines as well as it correlation.

** Yes - most of the data we have cited comes from work on humans and other mammals - although we try to be careful not to mislead the readers by stating this also applied to ectotherms. We have also cut sections in the Discussion that were repeated and think and hope it reads better now.

4. Line 179, correct the Typo, Forr

** Thanks - been corrected.

5. Line 290-291, does author means Asghar et al. 2018?

** No - as stated (2015)

6. line 296 - 297, Please add reference.

** Thanks, has been added.

7. Line 327-330. In recent study, it has been shown that although cell promotion change during infection but does not correlate with whole blood telomere length (Asghar et al. 2018).

** OK - but this is not really relevant to what is talked about in this paragraph (complications of comparisons across taxa).

Reviewer: 3

Comments to the Author(s)

This is a really interesting study on average telomere length differences in three blood cells types in reptiles: red blood cells, lymphocytes and azurophils. The authors use a study system, the painted dragon, in which they have plenty of experience and information, in addition to having developed a robust methodology for their aim.

I generally agree on their interpretation of the results, though I feel the authors should take more careful consideration when stressing the implications of their findings in the evolutionary context.

** We totally agree and were hoping to do that by including data from the literature. The problem is there is no data available (at all!) on e.g., life span of different 'white blood cells' in reptiles - only a few on fish - so it's hard to elaborate on evolutionary relationships, grounded in our own work, without quickly becoming speculative.

I agree on the importance of the extant differences in telomere length between cell types. However,

I feel the discussion of the authors on this respect could be more focus as I find it too loose for the brief of the results, i.e. differences on average telomere length among three blood cell types.

**As mentioned above, we have cut sections that were repeated in the discussion - but to the best of our knowledge this is the first time someone has discussed different parameters and processes that collectively affect telomere length in and across taxa - and how these changes with cell and whole-organism life. I think it's worth an extra page on that to encourage other researchers to collect similar data on different taxa.

Furthermore, as stated by the authors (L282-297), the proportion of each cell type should be critical to over/underestimate blood cell average telomere length. Indeed, in the study model, the longer average telomere length in the azurophils might not be relevant for the final telomere length of the individual. I wonder if the authors have information on the final telomere length estimate in the whole blood sample and in that case how the estimated within-cell telomere length influence this? I think adding this kind of information will provide a direct example for the point discussed in L282-297.

** This is a very good point but unfortunately our blood volume was insufficient to also allow for additional analyses of whole blood telomere length using other methods (e.g., TRF).

I understand that cell counts are not always easily normalised, but have the authors considered to analyse the telomere distribution of these three types in a single model instead (L184-187)? E.g. using the current DataBase in long-format with telomere length as a dependent variable, the cell type as a 3-level factor and the ID of the individual as a random effect to account for the non-independence in the measurements.

** No we had not done that before - good suggestion. However, it doesn't change much.

With raw data - we get these tests for normality for the pooled data set:

Tests for Normality				
Test	Statistic		p Value	
Shapiro-Wilk	W	0.903669	Pr < W	<0.0001
Kolmogorov-Smirnov	D	0.109559	Pr > D	<0.0100
Cramer-von Mises	W-Sq	0.360563	Pr > W-Sq	<0.0050
Anderson-Darling	A-Sq	2.73134	Pr > A-Sq	<0.0050

And this for the log-transformed data:

Tests for Normality				
Test	Statistic		p Value	
Shapiro-Wilk	W	0.963177	Pr < W	0.0037
Kolmogorov-Smirnov	D	0.093843	Pr > D	0.0174

Tests for Normality

Test	Statistic	p Value
Cramer-von Mises	W-Sq 0.236277	Pr > W-Sq <0.0050
Anderson-Darling	A-Sq 1.427546	Pr > A-Sq <0.0050

Running the mixed effect model gives:

Solution for Fixed Effects

Effect	celltype	Estimate	Standard Error	DF	t Value	Pr > t
Intercept		-1.4365	0.05940	107	-24.18	<.0001
celltype	az	0.3721	0.08078	72	4.61	<.0001
celltype	rb	-0.9980	0.08078	72	-12.36	<.0001
celltype	wb	0

Type 3 Tests of Fixed Effects

Effect	Num DF	Den DF	F Value	Pr > F
celltype	2	72	153.86	<.0001

So - it doesn't change any of the results and given the MS is quite long for its content already, I see no point in adding it to the already presented results.

Minor

L104-126. I suggest to directly provide in the text the information about the individuals used in the study. It is nice to have information about the species ecology but sex, age or sample size should be easily accessible, so far only available in the DataBase.

** This has now been added (l. 105-107)

L190-193. Could the authors be slightly more specific about the model selection process used? Also, the authors state that they used mixed models, in that case which is the random structure? Please state it.

** Terribly sorry - the models were not mixed - just generalized linear models (Proc GLM in SAS). This has been rephrased. We simply excluded the azurophils in one model to make the data comparable to similar for mammals and birds.

Figure 3. Please explain in the figure legend the x-axis or reword it. I find the current nomenclature confusing as it might be confused by the proportion of total cells. Also there is a potential typo, the two versions of the image have different axis title and scale.

** Thanks for picking up on this (relic from a previous figure version) - this has now been fixed and the order of the bars synchronized between figure legend and the bars in the diagram.